# OpenMatch: Open-set Consistency Regularization for Semi-supervised Learning with Outliers

**Kuniaki Saito**[1]  **Donghyun Kim**[1]  **Kate Saenko**[1,2]
[1]Boston University  [2]MIT-IBM Watson AI Lab

[`keisaito,donhk,saenko`]@bu.edu

## Abstract

Semi-supervised learning (SSL) is an effective means to leverage unlabeled data to improve a model's performance. Typical SSL methods like FixMatch assume that labeled and unlabeled data share the same label space. However, in practice, unlabeled data can contain categories unseen in the labeled set, *i.e.*, outliers, which can significantly harm the performance of SSL algorithms. To address this problem, we propose a novel Open-set Semi-Supervised Learning (OSSL) approach called *OpenMatch*. Learning representations of inliers while rejecting outliers is essential for the success of OSSL. To this end, OpenMatch unifies FixMatch with novelty detection based on one-vs-all (OVA) classifiers. The OVA-classifier outputs the confidence score of a sample being an inlier, providing a threshold to detect outliers. Another key contribution is an open-set soft-consistency regularization loss, which enhances the smoothness of the OVA-classifier with respect to input transformations and greatly improves outlier detection. OpenMatch achieves state-of-the-art performance on three datasets, and even outperforms a fully supervised model in detecting outliers unseen in unlabeled data on CIFAR10. The code is available at `https://github.com/VisionLearningGroup/OP_Match`.

## 1 Introduction

Semi-supervised learning (SSL) leverages unlabeled data to improve a model's performance [27, 38, 2, 1, 35, 22, 37]. An SSL model can propagate the class information of a small set of labeled data to a large set of unlabeled data, which significantly improves the recognition accuracy without any additional annotation cost. A common assumption of SSL is that the label spaces of labeled and unlabeled data are identical, but, in practice, the assumption is easily violated. Depending on how it was collected, the unlabeled data may contain novel categories unseen in the labeled training data, *i.e.*, outliers. Since these outliers can significantly harm the performance of SSL algorithms [14], detecting them is necessary to make SSL more practical. Ideally, a model should classify samples of known categories *i.e.*, inliers, into correct classes while identifying samples of novel categories as *outliers*. This task is called *Open-set Semi-supervised Learning* (OSSL) [44]. While OSSL is a more realistic and practical scenario than standard SSL, it has not been as widely explored.

Existing strong SSL methods [35, 41] do not work well for OSSL. For example, FixMatch [35] generates pseudo-labels using the model's predictions on weakly augmented unlabeled images and trains the model to match its predictions on strongly augmented images with the pseudo-labels. This method exploits the advantage of pseudo-labeling as well as regularizing a model with the consistency between differently augmented images. But, in OSSL, this risks assigning pseudo-labels of known categories to outliers, which degrades the recognition accuracy. A possible solution is to compute the SSL objective only for unlabeled samples considered to be inliers, where confidence thresholding is used to pick inliers. For instance, MTC [44] regards some proportion of samples as outliers by using Otsu thresholding [25]. However, this is not robust to varying proportions of outliers as discussed in

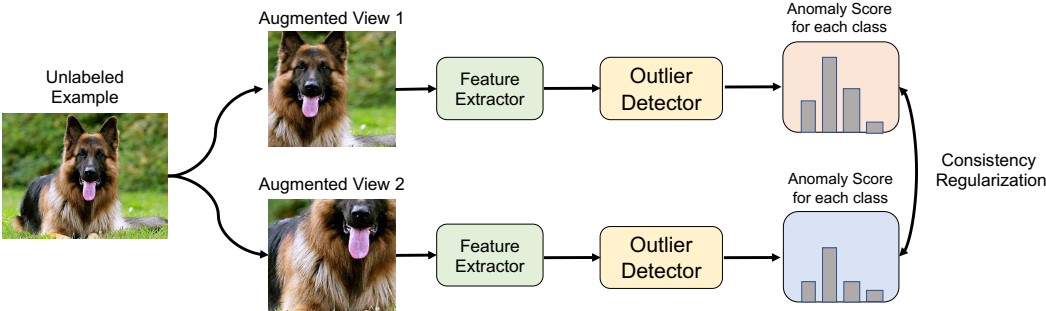

Figure 1: An illustration of our proposed open-set soft-consistency loss used to enhance outlier detection. Two differently augmented inputs are fed into the network to obtain the predictions of the outlier detector. The detector consists of one-vs-all classifiers and is able to detect outliers in an unsupervised way. The consistency loss is computed in a soft manner, *i.e.*, without sharpening logits.

[44]. DS3L [14] proposes meta-optimization that attempts to pick unlabeled data useful to improve generalization performance. But, this method does not have an objective to separate inliers from outliers.

Given the limitations of the existing methods, we aim to learn representations that separate outliers from inliers in a feature space and a threshold effective for detecting outliers. The challenge is learning such representations with a small number of labeled inliers and no supervision to find outliers. Moreover, choosing an accurate threshold value is not a trivial problem.

We propose a new framework, *OpenMatch*, to address the above drawbacks of OSSL. First, we propose to utilize a One-Vs-All (OVA) network [32] that can learn a threshold to distinguish outliers from inliers. A separate OVA-classifier is trained for each class, and a sample is labeled an outlier if all of the classifiers determine it to be one. Thus, this technique allows us to identify outliers in an unsupervised way. We call this an *outlier detector*. Second, we propose a novel *open-set soft-consistency loss* to learn more effective representations for detecting outliers. We first transform an unlabeled input in two ways and obtain two logits from the outlier detector. Then, we minimize the distance between the two logits to encourage consistency (See Fig. 1). The main difference between SSL and OSSL is that unlabeled outliers do not have any neighboring labeled samples, which makes it risky to perform hard-labeling such as pseudo-labeling. The outlier detector outputs a distance from inliers given an input, and enhancing the smoothness of this function allows us to improve its ability to find outliers. Empirically, this objective provides significant improvements in detecting outliers. Finally, to correctly classify inliers, we propose to apply FixMatch [35] to unlabeled samples that are considered to be inliers by the outlier detector.

The resulting framework shows consistent gains over baselines on various datasets and settings of OSSL. For example, OpenMatch achieves a 10.4% error rate on CIFAR10 with 300 labeled examples compared to the previous state-of-the-art of 20.3%. Surprisingly, OpenMatch demonstrates good performance in detecting outliers unseen in unlabeled training data. For instance, in the experiments on CIFAR10 with 100 labels per class, OpenMatch achieves a 3.4% higher AUROC in detecting outliers than a supervised model trained with all training samples. To summarize, our contributions are as follows:

- A soft open-set consistency regularization (SOCR), to improve outlier detection in OSSL.
- A new framework, OpenMatch, which combines a OVA-classifier, SOCR, and FixMatch.
- A new state-of-the-art in both correctly classifying inliers and detecting outliers, even when the outliers are unseen in unlabeled training data.

## 2    Related Work

**Semi-supervised Learning.** Pseudo-labeling (PL) is a strong baseline in semi-supervised learning [23], and methods like FixMatch [35] or UDA [41] combining data augmentation and PL show the highest performance on many benchmark datasets. However, even with these strong methods,

performance drops when a model is exposed to noisy unlabeled data that includes novel categories. On the other hand, methods based on soft consistency regularization enforce smoothness of the decision boundary with respect to stochastic transformations [34] or models [22, 37]. By "soft", we mean that no sharpening or pseudo-labeling is applied to logits to propagate training signals through soft targets. The soft consistency regularization can be understood as approximated Jacobian regularization method [12, 31]. We propose to apply soft consistency regularization to our outlier detector, which we call soft open-set consistency regularization. Training a model with self-supervised learning and SSL achieves label-efficient training [5, 46]. Self-supervised learning and SSL seem to be complementary to each other to improve the recognition accuracy. But, unlabeled data with outliers will hurt the performance of these approaches since they rely on existing SSL objectives. Since our work focuses on the aspect of SSL, we hope it is complementary to self-supervised learning.

**Open-set SSL** methods include MTC [44], D3SL [14], and UASD [6]. MTC updates the network parameters and the anomaly score of unlabeled data alternately. In addition, it minimizes SSL loss (MixMatch [2] for a part of unlabeled data that is considered to be inliers. D3SL selectively employs unlabeled data to optimize SSL loss as well as optimizing a function that selects unlabeled data. UASD generates soft targets of the classifier for inliers (closed-set classifier) by averaging over many temporally ensembled networks' predictions. By contrast, OpenMatch does not have to store the temporal ensembles and applies soft consistency loss to the *outlier detector*. Our empirical results show that the consistency loss for the outlier detector outperforms that for the closed-set classifier by a lot (See Sec. 4.4).

**Open-set Domain Adaptation (ODA).** OSSL is similar to open-set domain adaptation [3, 33] in that the unlabeled data contains novel categories. A key difference is that the unlabeled and labeled data follow different data distributions. Another key difference is that we aim to train a model from scratch while the domain adaptation task assumes access to models pre-trained on ImageNet [9]. Since the ImageNet contains 1,000 diverse categories, the pre-trained models have discriminative representations useful to detect outliers. In this point, our task is more challenging than ODA. To detect outliers in an unsupervised way, we employ a OVA-classifier model [32] proposed for this task.

**Novelty Detection.** Novelty detection or open-set classification aims to identify outliers that are completely unseen during training [17, 24, 10, 11]. Self-supervised learning such as rotation prediction and contrastive learning is shown to be useful to separate outliers from inliers [19, 36, 40]. The task assumes plenty of labeled inliers in training data. Padhy *et al.* employ OVA-classifiers for this task [30]. Hendrycks *et al.* reveal that exposing a model to outlier data allows it to effectively detect anomalies [18] and train it on out-of-distribution training data in a supervised way. But, collecting such useful out-of-distribution data is not always possible. In OSSL, a model cannot be trained on many labeled samples and needs to detect which samples are inlier or outliers in unlabeled data. In this sense, OSSL is more challenging and more realistic.

## 3 OpenMatch

**Problem Setting.** Our task is to learn a classifier using open-set semi-supervised learning. For a $K$-way classification problem, let $\mathcal{X} = \{(x_b, y_b) : b \in (1, ...B)\}$ be a batch of $B$ labeled examples $x_b$ randomly sampled from the labeled training set, $\mathcal{S}$, where $y_b \in (1, ...K)$ is the corresponding label. Let $\mathcal{U} = \{(u_b) : b \in (1, ...\mu B)\}$ be a batch of $\mu B$ data randomly sampled from the unlabeled training set, $\mathcal{S}_u$, where $\mu$ is a hyper-parameter that determines the relative sizes of $\mathcal{X}$ and $\mathcal{U}$. Unlike in SSL, which assumes all unlabeled data come from the $K$ known classes, in our OSSL setting they may come from unknown outlier classes. The goal of OSSL is to classify inliers into the correct classes while learning to detect the outliers.

**Approach Overview.** Our model has three components: (1) a shared feature extractor $F(\cdot)$, (2) an outlier detector consisting of $K$ one-vs-all (OVA) sub-classifiers $D^j(\cdot), j \in \{1, ..., K\}$, and (3) a closed-set classifier $C(\cdot)$ which outputs a probability vector $\in \mathbb{R}^K$ for $K$-class classification. At test time, the closed-set classifier is first applied to predict the $K$-way label, $\hat{y}$. If $D^{\hat{y}}$ predicts an inlier, then the output class is $\hat{y}$, otherwise, the output is "outlier". The main technical novelty comes from the choice of OVA-classifiers for outlier detection as well as training them with soft open-set consistency regularization. We first describe the training of the OVA-classifiers [32] before describing the remaining details of our framework.

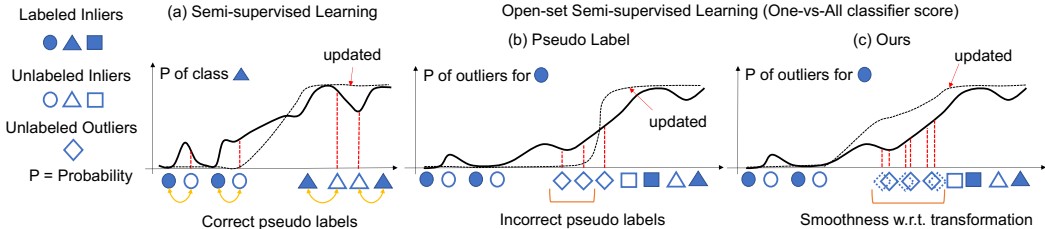

Figure 2: **(a) SSL with pseudo-labels.** Hard labels such as pseudo-labels are often used in SSL to propagate training signals from labeled samples to neighboring unlabeled ones. **(b) OSSL with pseudo-labels.** However, in OSSL, the outlier unlabeled samples do not have any labeled neighbors. Therefore, the pseudo-labels can be highly unreliable. **(c) OSSL with soft consistency (ours).** To address this, we propose to ensure smoothness with respect to data augmentation by minimizing a soft consistency loss. This separates the outliers from inliers and avoids using incorrect pseudo-labels.

## 3.1 One-vs-All Outlier Detector

For open-set [33] or universal domain adaptation [42], Saito *et al.* [32] propose to train OVA-classifiers to detect outlier samples of an unlabeled target domain. They aim to learn a boundary between inliers and outliers for each class. Each sub-classifier is trained to distinguish if the sample is an inlier for the corresponding class. For example, the sub-classifier for class $j$, $D^j$, outputs a 2-dimensional vector $z_b^j = D^j(F(x_b)) \in \mathbb{R}^2$, where each dimension indicates the score of a sample being an inlier and outlier respectively. We denote $p^j(t = 0|x_b)$ and $p^j(t = 1|x_b)$ as the probability of $x_b$ being the inlier and outlier for the class $j$, computed by Softmax($z_b^j$) (Note that $p^j(t = 0|x_b) + p^j(t = 1|x_b) = 1$). Then, the sub-classifier $D^j$ is trained such that the data from class $j$ is treated as positives, and the data from all the other classes is treated as negatives. To achieve this, we minimize the following loss for the outlier detector for a given batch $\mathcal{X} := \{x_b, y_b\}_{b=1}^B$:

$$\mathcal{L}_{ova}(\mathcal{X}) := \frac{1}{B} \sum_{b=1}^B -\log(p^{y_b}(t = 0|x_b)) - \min_{i \neq y_b} \log(p^i(t = 1|x_b)). \quad (1)$$

Here we use the hard-negative sub-classifier sampling technique proposed by [32] to effectively learn the threshold. We refer the reader to [32] for more details. The key is that each sub-classifier outputs a distance representing how far the input is from the corresponding class. Therefore, the classifiers are effective to identify unlabeled outliers. The output of $D^j$ is used to detect outliers as follows. If the probability of a sample being an outlier for the class highest-scored by $C()$ is larger than that of a sample being an inlier, we consider the input as an outlier. To be specific, we identify the unlabeled sample $u_b$ as an outlier if $p^{\hat{y}}(t = 0|u_b) < 0.5$, where $\hat{y} = \arg\max_j C(F(u_b))$. This detection is employed in selecting pseudo-inliers from unlabeled data as explained in Sec. 3.3.

**Open-set Entropy Minimization.** For unlabeled samples, [32] apply entropy minimization with respect to the OVA-classifiers, denoted by $L_{em}(\mathcal{U})$. The separation between inliers and outliers is enhanced through the minimization of this loss for a given batch $\mathcal{U} := \{u_b\}_{b=1}^{\mu B}$:

$$\mathcal{L}_{em}(\mathcal{U}) := -\frac{1}{\mu B} \sum_{b=1}^{\mu B} \sum_{j=1}^K \quad p^j(t = 0|u_b) \log p^j(t = 0|u_b) + p^j(t = 1|u_b) \log p^j(t = 1|u_b). \quad (2)$$

The difference between our setup and [32] is that they employ a model pre-trained on ImageNet [9], which extracts highly discriminative features, thus, inliers and outliers are well-separated even before fine-tuning. However, we aim to train a model to learn representations separating them from scratch. Thus the inliers and outliers are confused before training, and we need an objective to ensure the separation in addition to Eq. 2.

## 3.2 Soft Open-set Consistency Regularization (SOCR)

To motivate our approach, we describe the difference between SSL and OSSL in Fig. 2. In SSL, the hard consistency loss such as pseudo-labeling can be effective to propagate label information from

**Algorithm 1** OpenMatch Algorithm.

---

1: **Input:** Set of labeled data $\mathcal{S} = \big((x_b, y_b); b \in (1, \dots, N)\big)$, set of unlabeled data $\mathcal{S}_u = \big(u_b; b \in (1, \dots, N_u)\big)$, and set of pseudo-inlier data $\mathcal{K} = \emptyset$.
Data augmentation function $\mathcal{T}$. Model parameters $w$, learning rate $\eta$, epoch $E_{fix}$ and $E_{\max}$, iteration $I_{\max}$, trade-off parameters, $\lambda_{em}, \lambda_{oc}, \lambda_{fm}$;
**for** *Epoch = 1 to $E_{\max}$* **do**
    **for** *Iteration = 1 to $I_{\max}$* **do**
        2: **Sample** a batch of labeled data $\mathcal{X} \in \mathcal{S}$ and unlabeled data $\mathcal{U} \in \mathcal{S}_u$;
        3: **Compute** $\mathcal{L}_{all} = \mathcal{L}_{sup}(\mathcal{X}) + \lambda_{em}\mathcal{L}_{em}(\mathcal{U}) + \lambda_{oc}\mathcal{L}_{oc}(\mathcal{U}, \mathcal{T})$;        // Eq.1, 2 and 3
        **if** *Epoch > $E_{fix}$* **then**
            4: **Sample** a batch of pseudo-inliers $\mathcal{I} \in \mathcal{K}$;         // Sample pseudo-inliers.
            5: **Compute** $\mathcal{L}_{all} \mathrel{+}= \lambda_{fm}\mathcal{L}_{fm}(\mathcal{I})$ ;         // FixMatch for pseudo-inliers.
        **end**
        6: **Update** $w = w - \eta\nabla_w L_{all}$;         // Update weights
    **end**
    **if** *Epoch $\geq E_{fix}$* **then**
        7: **Update** $\mathcal{K} = \mathrm{Select}(w, \mathcal{D}_u)$;         // Detect outliers and select pseudo-inliers.
    **end**
**end**
8: **Output:** Model parameters $w$.

---

labeled to unlabeled samples. Through the hard consistency loss, unlabeled samples get training signals from neighboring labeled samples (Fig. 2(a)). By contrast, in OSSL, since outliers are not assigned any labels, labeled samples are far from outliers, which makes the hard labeling unreliable, and unsupervised training signals can be incorrect (Fig. 2(b)). The open-set entropy minimization above can also be considered a variant of the hard consistency since it is minimized if the predicted output class distribution is one-hot. To propagate useful training signals to unlabeled samples, we propose to smooth the decision boundary of the outlier detector by minimizing the distance between its predictions on two augmentations of the same image (Fig. 2(c)). We use unsharpened logits, keeping the training signal soft to avoid introducing incorrect pseudo-labels. We call this loss *soft open-set consistency regularization (SOCR)*. Given the smoothed outlier detector, the hope is that the training signals of the open-set entropy minimization will also become more correct.

Specifically, SOCR enhances the smoothness of the outlier detector over data augmentation $\mathcal{T}$, which in our experiments is just standard random cropping. Note that the standard cropping is also used in computing Eq. 1 and 2, but we omit it for the simplicity of notation. We obtain two different views of $u_b$, $\mathcal{T}_1(u_b)$ and $\mathcal{T}_2(u_b)$, where $\mathcal{T}_1$ and $\mathcal{T}_2$ are data augmentation functions stochastically sampled from $\mathcal{T}$. Let $p^j(t|\mathcal{T}_1(u_b))$ be the output logits of $\mathcal{T}_1(u_b)$ from the $j$th OVA-classifier. We encourage the consistency of the output logits over $\mathcal{T}$ to enhance the smoothness by minimizing the following loss for a given batch $\mathcal{U} := \{u_b\}_{b=1}^{\mu B}$:

$$\mathcal{L}_{oc}(\mathcal{U}, \mathcal{T}) := \frac{1}{\mu B} \sum_{b=1}^{\mu B} \sum_{j=1}^{K} \sum_{t \in (0,1)} |p^j(t|\mathcal{T}_1(u_b)) - p^j(t|\mathcal{T}_2(u_b))|^2. \tag{3}$$

This minimizes the distance between two probability outputs to ensure smoothness with respect to $\mathcal{T}$. Note that the key part of the regularization is not applying any sharpening to the output logits, keeping them soft. The major difference from the previous soft consistency regularization [12, 37, 22] is that we use the outlier detector to compute the regularization loss.

### 3.3 Overall Framework

The entire OpenMatch algorithm is described in Alg. 1. The unsupervised losses (Eqs. 2, 3) for the outlier detector are effective to detect outliers in unlabeled data. However, these losses are not sufficient to correctly classify unlabeled inliers. Therefore, we propose to introduce a semi-supervised learning loss for unlabeled samples classified as inliers. We adopt FixMatch [35] since it is a simple, yet very strong SSL method. FixMatch first generates pseudo-labels using the model's predictions on weakly augmented unlabeled images. The model is then trained to predict these pseudo-labels when fed a strongly augmented version of the same image. "Weak" denotes augmentations such as a simple

random cropping while "strong" refers to extensive data augmentation such as RandAugment [8] and CTAugment [1].

For labeled samples, we also compute the standard cross-entropy loss to train $C(\cdot)$, $\mathcal{L}_{cls}(\mathcal{X})$, for the closed-set output, $C(F(x_b))$. We summarize $\mathcal{L}_{sup}(\mathcal{X})$ as the sum of $\mathcal{L}_{ova}(\mathcal{X})$ and $\mathcal{L}_{cls}(\mathcal{X})$. After training a model for $E_{fix}$ epochs with $\mathcal{L}_{sup}, \mathcal{L}_{em}, \mathcal{L}_{oc}$, we begin to select pseudo-inliers from unlabeled data at every following epoch. Specifically, in Alg. 1, $\text{Select}(w, \mathcal{D}_u)$ denotes the process of classifying unlabeled samples given model parameters $w$ of models ($F(\cdot), D(\cdot),$ and $C(\cdot)$), and unlabeled data. The outlier-detection process is summarized in Sec. 3.1.

Then, the FixMatch loss, $\mathcal{L}_{fm}(\mathcal{I})$, is computed only for the pseudo-inliers to optimize the model, where $\mathcal{I}$ denotes a batch of pseudo-inliers. The following is the resulting overall objective of OpenMatch:

$$\mathcal{L}_{all}(\mathcal{X}, \mathcal{U}, \mathcal{T}, \mathcal{I}) := \mathcal{L}_{sup}(\mathcal{X}) + \lambda_{em}\mathcal{L}_{em}(\mathcal{X}) + \lambda_{oc}\mathcal{L}_{oc}(\mathcal{U}, \mathcal{T}) + \lambda_{fm}\mathcal{L}_{fm}(\mathcal{I}), \quad (4)$$

where $\lambda_{em}, \lambda_{oc}$ and $\lambda_{fm}$ control the trade-off for each objective.

# 4 Experiments

**Setup.** We evaluate the efficacy of OpenMatch on several SSL image classification benchmarks. Specifically, we perform experiments with varying amounts of labeled data and varying numbers of known/unknown classes on CIFAR10/100 [21] and ImageNet [9]. We attempt to cover various scenarios by using these three datasets.

Note that we use an identical set of hyper-parameters except for $\lambda_{oc}$, which is tuned on each dataset. $\lambda_{em}$ is set 0.1 in all experiments. $\lambda_{fm}$ is set to 0 before $E_{fix}$ epochs and then set to 1 for all experiments. $E_{fix}$ is set to 10 in all experiments. The hyper-parameters for FixMatch, e.g., data augmentation, confidence threshold, are fixed across all experiments for simplicity. The hyper-parameters are set by tuning on a validation set that contains a small number of labeled samples. Note that the validation set does not contain any outliers. A complete list of hyper-parameters is reported in the appendix. Each experiment is done with a single 12-GB GPU, such as an NVIDIA TitanX.

**Baselines.** For the OSSL baseline, we employ MTC [44] using the author's implementation. In addition, we show the result of a model trained only with labeled samples (Labeled Only). We exclude UASD [6] and D3SL [14] from our baselines since their reported results underperform a model we train only with labeled samples. For instance, in the experiments for CIFAR10 at 400 labels per class, their reported error rates are worse than 20% while the rate of the Labeled Only baseline is 20.0%. A comparison with FixMatch [35] reveals the effect of OVA-classifiers trained with the soft consistency loss. The hyper-parameters of both OpenMatch and baselines are tuned by maximizing the accuracy on the validation set. Since the validation set does not have any outliers, we choose the validation accuracy as a criterion.

**Evaluation.** We assume the test set contains both known (inlier) and unknown (outlier) classes. For known classes, classification accuracy is used to evaluate performance. To evaluate the separation into inliers and outliers, we use AUROC following the standard evaluation protocol of novelty detection [17]. To compute the anomaly score, the outlier detector's score is employed for OpenMatch, Labeled Only, and FixMatch, where we add our outlier detector to the latter two models. We report the results averaged over three runs and their standard deviations.

## 4.1 CIFAR10 and CIFAR100

We compare OpenMatch with several baselines on the standard benchmark datasets of SSL, CIFAR10, and CIFAR100. Following [44], we use a randomly initialized Wide ResNet-28-2 [45] with 1.5M parameters in these experiments. For CIFAR10, we split the classes into known and unknown classes by defining animal classes as known and others as unknown, which results in 6 known classes and 4 unknown classes. For CIFAR100, we first split the classes by their super-classes provided by the dataset so that known and unknown classes do not share super-classes. Then, we split the super-classes into known and unknown. To evaluate performance with different numbers of outliers, we run experiments on two settings: 80 known classes (20 unknown classes) and 55 known classes (45 unknown classes).

Tables 1 and 2 describe the error rate on inliers and AUROC values respectively. Note that the number of labeled samples *per class* is shown in each column. With respect to the error rate, OpenMatch achieves state-of-the-art performance in all cases, and the gain over the baselines is remarkable in

| Dataset | CIFAR10 | | | CIFAR100 | | CIFAR100 | | ImageNet-30 |
|---|---|---|---|---|---|---|---|---|
| No. of Known / Unknown | 6 / 4 | | | 55 / 45 | | 80 / 20 | | 20 / 10 |
| No. of labeled samples | 50 | 100 | 400 | 50 | 100 | 50 | 100 | 10 % |
| Labeled Only | $35.7_{\pm1.1}$ | $30.5_{\pm0.7}$ | $20.0_{\pm0.3}$ | $37.0_{\pm0.8}$ | $27.3_{\pm0.5}$ | $43.6_{\pm0.5}$ | $34.7_{\pm0.4}$ | $20.9_{\pm1.0}$ |
| FixMatch [35] | $43.2_{\pm1.2}$ | $29.8_{\pm0.6}$ | $16.3_{\pm0.5}$ | $35.4_{\pm0.7}$ | $27.3_{\pm0.8}$ | $41.2_{\pm0.7}$ | $34.1_{\pm0.4}$ | $12.9_{\pm0.4}$ |
| MTC [44] | $20.3_{\pm0.9}$ | $13.7_{\pm0.9}$ | $9.0_{\pm0.5}$ | $33.5_{\pm1.2}$ | $27.9_{\pm0.5}$ | $40.1_{\pm0.8}$ | $33.6_{\pm0.3}$ | $13.6_{\pm0.7}$ |
| OpenMatch | $\mathbf{10.4_{\pm0.9}}$ | $\mathbf{7.1_{\pm0.5}}$ | $\mathbf{5.9_{\pm0.5}}$ | $\mathbf{27.7_{\pm0.4}}$ | $\mathbf{24.1_{\pm0.6}}$ | $\mathbf{33.4_{\pm0.2}}$ | $\mathbf{29.5_{\pm0.3}}$ | $\mathbf{10.4_{\pm1.0}}$ |

Table 1: Error rates (%) with standard deviation for CIFAR10, CIFAR100 on 3 different folds. Lower is better. For ImageNet, we use the same fold and report averaged results of three runs. Note that the number of labeled samples *per class* is shown in each column.

| Dataset | CIFAR10 | | | CIFAR100 | | CIFAR100 | | ImageNet-30 |
|---|---|---|---|---|---|---|---|---|
| No. of Known / Unknown | 6 / 4 | | | 55 / 45 | | 80 / 20 | | 20 / 10 |
| No. of labeled samples | 50 | 100 | 400 | 50 | 100 | 50 | 100 | 10 % |
| Labeled Only | $63.9_{\pm0.5}$ | $64.7_{\pm0.5}$ | $76.8_{\pm0.4}$ | $76.6_{\pm0.9}$ | $79.9_{\pm0.9}$ | $70.3_{\pm0.5}$ | $73.9_{\pm0.9}$ | $80.3_{\pm1.0}$ |
| FixMatch [35] | $56.1_{\pm0.6}$ | $60.4_{\pm0.4}$ | $71.8_{\pm0.4}$ | $72.0_{\pm1.3}$ | $75.8_{\pm1.2}$ | $64.3_{\pm1.0}$ | $66.1_{\pm0.5}$ | $88.6_{\pm0.5}$ |
| MTC [44] | $96.6_{\pm0.6}$ | $98.2_{\pm0.3}$ | $98.9_{\pm0.1}$ | $81.2_{\pm3.4}$ | $80.7_{\pm4.6}$ | $79.4_{\pm2.5}$ | $73.2_{\pm3.5}$ | $93.8_{\pm0.8}$ |
| OpenMatch | $\mathbf{99.3_{\pm0.3}}$ | $\mathbf{99.7_{\pm0.2}}$ | $\mathbf{99.3_{\pm0.2}}$ | $\mathbf{87.0_{\pm1.1}}$ | $\mathbf{86.5_{\pm2.1}}$ | $\mathbf{86.2_{\pm0.6}}$ | $\mathbf{86.8_{\pm1.4}}$ | $\mathbf{96.4_{\pm0.7}}$ |

Table 2: AUROC of Table 1. Higher is better. Note that the number of labeled samples *per class* is shown in each column.

many cases. In particular, we see about a 10% improvement in CIFAR10 error at 50 labels. In addition, the improvements in AUROC are also clear. MTC [44] does not improve the performance very much compared to the labeled-only model on CIFAR100 because their method of identifying outliers, *i.e.*, Otsu's thresholding [25], is not very robust to the number of outliers as discussed in their paper. On the other hand, OpenMatch improves the error rate by more than 3% and AUROC by more than 6% in all cases. Simply applying FixMatch [35] also does not always improve the error compared to the labeled-only model. This is because FixMatch confuses inliers and outliers and assigns pseudo-labels to outliers, as the significant decrease in AUROC indicates.

## 4.2 ImageNet

We evaluate OpenMatch on ImageNet to see its behavior on more complex and challenging datasets. Since training on the entire ImageNet is computationally expensive, we choose ImageNet-30 [17], which is a subset of ImageNet containing 30 classes. The subset is useful for evaluation on unseen outliers as done in Sec. 4.5, because the classes do not include several super-classes such as a bird class. We pick the first 20 classes, in alphabetical order, as known classes, and the remaining 10 classes as unknown. We utilize the ResNet-18 [16] network architecture. Hyper-parameters such as a batch-size, learning rate, data augmentation are the same as ones used in CIFAR10 and CIFAR100. As shown in the rightmost columns of Tables 1 and 2, OpenMatch achieves the best performance in both accuracy and AUROC. Unlike the results on CIFAR, FixMatch improves the performance over the Labeled Only model. This is probably because outliers are easier to detect, as the AUROC score of the Labeled Only model already shows a higher score than other scenarios. Therefore, even without any outlier detection objectives, FixMatch can correctly select pseudo-inliers with confidence thresholding.

## 4.3 Ablation Study on Soft Open-set Consistency Regularization

In this section, we validate our claim that the SOCR regularization is effective at separating inliers and outliers. We perform several ablation studies to see the benefit of SOCR and measure the model's ability to detect outliers using AUROC. In summary, both numerical and qualitative evaluations confirm the effectiveness of SOCR.

Table 3 describes the numerical comparison of ablated models. In this evaluation, we do not apply FixMatch to pseudo-inliers to measure the pure gain from the consistency loss. Introducing SOCR increases AUROC by more than 20% in CIFAR10, 8% in CIFAR100, and 8% in ImageNet. These results support the effectiveness of smoothing the outlier detector's output.

| Dataset | CIFAR10 | | CIFAR100 | | ImageNet-30 |
|---|---|---|---|---|---|
| No. Known / Unknown | 6 / 4 | | 80 / 20 | | 20 / 10 |
| No. Labeled samples | 50 | 400 | 50 | 100 | 10 % |
| without SOCR | $60.5_{\pm 2.8}$ | $75.8_{\pm 0.8}$ | $70.4_{\pm 0.1}$ | $73.2_{\pm 0.2}$ | $81.3_{\pm 0.4}$ |
| with SOCR | $\mathbf{81.3}_{\pm 2.9}$ | $\mathbf{96.8}_{\pm 0.6}$ | $\mathbf{78.9}_{\pm 0.1}$ | $\mathbf{85.0}_{\pm 0.8}$ | $\mathbf{89.3}_{\pm 0.3}$ |

Table 3: Ablation study of our soft consistency regularization (SOCR, $\mathcal{L}_{oc}$). We report AUROC scores (%). In this study, we do not apply FixMatch to pseudo-inliers to see the pure gain from SOCR.

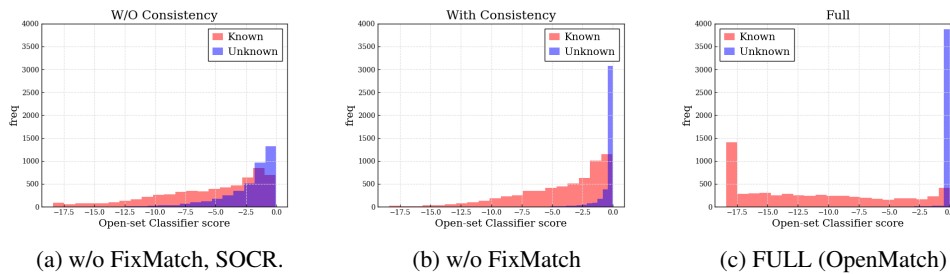

(a) w/o FixMatch, SOCR.  (b) w/o FixMatch  (c) FULL (OpenMatch)

Figure 3: The histograms of the outlier detector's scores obtained with ablated models. Red: Inliers, Blue: Outliers. From left to right, a model without FixMatch and SOCR, a model without FixMatch, and a model with all objectives. These results show that SOCR ensures separation between inliers and outliers, and FixMatch added to SOCR can further enhance this separation.

**Histograms of Anomaly Scores.** Fig. 3 illustrates the anomaly score of inlier and outlier test samples in the ablation experiment on CIFAR10. If SOCR is not employed (Fig. 3(a)), inliers and outliers are confused while SOCR makes the scores of outliers larger and enhances the separation (Fig. 3(b)). Further applying FixMatch to pseudo-inliers decreases the score of inliers (Fig. 3(c)), which separates inliers and outliers better. Considering the results of Table 2, just applying FixMatch to all unlabeled samples is often not useful to detect outliers, but applying it to the pseudo-inliers can improve the outlier detection.

## 4.4 Analysis

**Feature Visualization.** Fig. 4 visualizes the feature distributions with T-SNE [26] in the experiment on CIFAR10 at 100 labels. OpenMatch (c) separates inliers and outliers well while a model trained only with 100 labeled samples (a) confuses them. The separation of OpenMatch appears to be better than a model trained with all labeled inliers (b).

**Closed-set Consistency vs Open-set Consistency.** The soft consistency loss is computed for the outlier detector in OpenMatch. An alternative is to compute it for the closed-set classifier as done in UASD [6]. We replace SOCR with the soft consistency loss with respect to the closed-set classifier and perform the same analysis as in Sec. 4.3. In CIFAR10 at 50 labeled samples, this model achieves $70.6 \pm 1.3$ % AUROC, which is 10.1 % better than a model without consistency loss and 10.7 % worse than a model with SOCR. In CIFAR10 at 400 labeled samples, this model achieves $88.5 \pm 2.4$ % AUROC, which is 12.7 % better than a model without consistency loss and 8.3 % worse than a model with SOCR. These results indicate that even using soft consistency for the closed-set classifier helps to detect outliers, as shown in UASD [6], but SOCR helps much more. Thus, our choice of combining both the outlier detector and soft consistency loss is validated in this analysis.

**Comparison with OOD with self-supervised learning.** We compare our method with outlier detection methods using self-supervised learning. Several works [19, 36] have shown that training a model with self-supervised learning objectives such as contrastive loss or rotation prediction loss improves OOD performance. We provide a brief analysis of these baselines in open-set SSL.

**Does self-supervised pre-training help open-set SSL?** A model trained with self-supervised learning objectives shows great performance in many downstream image recognition tasks. We investigate whether initializing a model trained with self-supervised learning (SimCLR [4]) improves perfor-

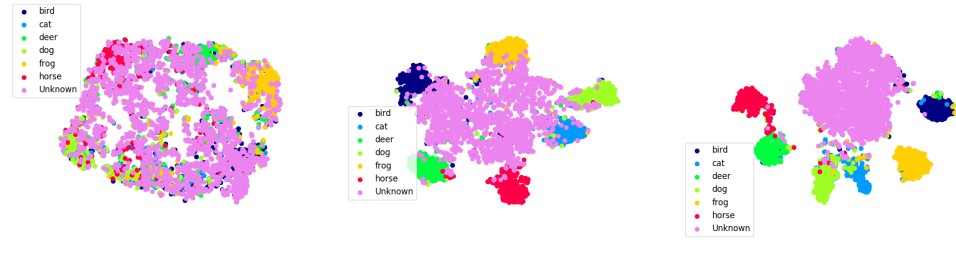

| (a) Partially labeled model | (b) Fully labeled model | (c) OpenMatch |

Figure 4: Feature visualization with t-SNE [26]. Different colors indicate different classes. Pink dots represent outliers. (a): A model trained with 100 labeled samples per class. (b): A model trained with all labeled inliers. (c): A model trained with OpenMatch. (Best viewed in color)

| Initialization | CIFAR10-50 | CIFAR10-400 | CIFAR100-50 |
|---|---|---|---|
| SimCLR Init | 10.6 / 99.3 | 5.9 / 99.4 | 27.1 / 87.1 |
| Random Init | 10.4 / 99.3 | 5.9 / 99.3 | 27.7 / 87.0 |

Table 4: Comparison by different initialization (Error (%) / AUROC). Initializing a model trained with SimCLR [4] for all unlabeled samples does not make a clear difference from random initialization.

mance in open-set SSL (Table 4). We pre-train a model with SimCLR loss using all unlabeled data. The initialization does not make a clear difference from random initialization, *i.e.*, default initialization.

**Does self-supervised training help open-set SSL?** Next, we investigate whether using self-supervised learning loss in addition to the classification loss on labeled data helps open-set SSL. In Table 5, we show the results of adding self-supervised loss for labeled data or unlabeled data. SimCLR does not help in any case compared to a model trained only with classification loss for labeled data. Constrative learning basically attempts to uniformly distribute all samples in a unit sphere, which does not necessarily help in open-set SSL. The goal of open-set SSL is to keep outliers far from inliers whereas clustering inliers within each class. By contrast, rotation prediction improves performance if applied to labeled samples in CIFAR-10. If applied to unlabeled data, a model can assign the same rotation labels to both inliers and outliers, which can prevent from separating them.

### 4.5 Novelty Detection

We have seen that OpenMatch can learn to detect outliers from unlabeled data. In this experiment, we evaluate how well our OSSL model can separate inliers from outliers unseen in the unlabeled data. This setting is similar to general outlier or out-of-distribution sample detection. Note that the models evaluated in this section are the same as the ones in Tables 1 and 2. We consider the following datasets as out-of-distribution: SVHN [28], resized LSUN [43], ImageNet, and CIFAR100 for CIFAR10 experiments (Table 6(a)), and LSUN, CUB-200 [39], Dogs [20], Caltech [13], Flowers [29], and DTD [7] for ImageNet-30 (Table 6(b)). See supplemental material for the details. We utilize AUROC to measure how well inliers and outliers are separated. To see the gap from a supervised model, we train a model using all labeled samples of inlier classes and show the comparison.

As seen in Table 6, OpenMatch outperforms OSSL baselines by 5.8% on average and even the supervised model by 3.4% on average on CIFAR10. On ImageNet-30, OpenMatch outperforms OSSL baselines by more than 4.1% on average. This result verifies that OpenMatch is robust to various out-of-distribution data by virtue of being exposed to unlabeled data containing outliers.

## 5 Conclusion

In this paper, we introduce a method for open-set semi-supervised learning (OSSL), where samples of novel categories are present in unlabeled data. To approach the task, we propose a novel framework, OpenMatch, which unifies one-vs-all classifiers and FixMatch. Our proposed objective, open-set soft consistency loss, is shown to be effective to detect outliers from unlabeled data, which allows FixMatch to work well in the OSSL setting. In addition, OpenMatch sometimes detects outliers unseen in unlabeled data better than a supervised model. We believe that our framework for OSSL will make label-efficient techniques more practical.

| Method | CIFAR10-50 | CIFAR100-50 |
|---|---|---|
| Labeled only | 63.9 $\pm$0.5 | 70.3 $\pm$0.5 |
| SimCLR: labeled samples | 63.5$\pm$0.7 | 68.6 $\pm$1.1 |
| SimCLR: unlabeled samples | 62.1$\pm$1.2 | 68.8$\pm$0.9 |
| Rot Pred: labeled samples | 70.0 $\pm$1.0 | 67.8 $\pm$0.5 |
| Rot Pred: unlabeled samples | 64.0 $\pm$1.3 | 67.0 $\pm$1.5 |
| SOCR | **81.3$\pm$2.9** | **78.9 $\pm$0.1** |

Table 5: Study about self-supervised learning loss for labeled or unlabeled data (AUROC). Rot Pred indicates the rotation prediction loss. SimCLR [4] either for labeled or unlabeled data slightly decreases the performance to detect outliers whereas rotation prediction for labeled data shows some gain in CIFAR-10.

| Method | CIFAR10 | Unseen Out-liers | | | | |
|---|---|---|---|---|---|---|
| | | SVHN | LSUN | ImageNet | CIFAR100 | MEAN |
| Labeled Only | 64.7$\pm$1.0 | 83.6$\pm$1.0 | 78.9$\pm$0.9 | 80.5$\pm$0.8 | 80.4$\pm$0.5 | 80.8$\pm$0.8 |
| FixMatch [35] | 60.4$\pm$0.4 | 79.9$\pm$1.0 | 67.7$\pm$2.0 | 76.9$\pm$1.1 | 71.3$\pm$1.1 | 73.9$\pm$1.3 |
| MTC [44] | 98.2$\pm$0.3 | 87.6$\pm$0.5 | 82.8$\pm$0.6 | 96.5$\pm$0.1 | 90.0$\pm$0.3 | 89.2$\pm$0.4 |
| OpenMatch | **99.7$\pm$0.1** | **93.0$\pm$0.4** | **92.7$\pm$0.3** | **98.7$\pm$0.1** | **95.8$\pm$0.4** | **95.0$\pm$0.3** |
| Supervised | 89.4$\pm$1.0 | 95.6$\pm$0.5 | 89.5$\pm$0.7 | 90.8$\pm$0.4 | 90.4$\pm$1.0 | 91.6$\pm$0.6 |

(a) Model trained on CIFAR10 (100 labeled data per class and unlabeled data.)

| Method | ImageNet-30 | Unseen Out-liers | | | | | | |
|---|---|---|---|---|---|---|---|---|
| | | LSUN | DTD | CUB | Flowers | Caltech | Dogs | MEAN |
| Labeled Only | 80.3$\pm$0.5 | 85.9$\pm$1.4 | 75.4$\pm$1.0 | 77.9$\pm$0.8 | 69.0$\pm$1.5 | 78.7$\pm$0.8 | 84.8$\pm$1.0 | 78.6$\pm$1.1 |
| FixMatch [35] | 88.6$\pm$0.5 | 85.7$\pm$0.1 | 83.1$\pm$2.5 | 81.0$\pm$4.8 | **81.9$\pm$1.1** | 83.1$\pm$3.4 | 86.4$\pm$3.2 | 83.0$\pm$1.9 |
| MTC [44] | 93.8$\pm$0.8 | 78.0$\pm$1.0 | 59.5$\pm$1.5 | 72.2$\pm$0.9 | 76.4 $\pm$2.1 | 80.9$\pm$0.9 | 78.0$\pm$0.8 | 74.2$\pm$1.2 |
| OpenMatch | **96.3$\pm$0.7** | **89.9$\pm$1.9** | **84.4$\pm$0.5** | **87.7$\pm$1.0** | 80.8$\pm$1.9 | **87.7$\pm$0.9** | **92.1$\pm$0.4** | **87.1$\pm$1.1** |
| Supervised | 92.8$\pm$0.8 | 94.4$\pm$0.5 | 92.7$\pm$0.4 | 91.5$\pm$0.9 | 88.2$\pm$1.0 | 89.9$\pm$0.5 | 92.3$\pm$0.8 | 91.3$\pm$0.7 |

(b) Model trained on ImageNet-30 (10 % of labeled data and unlabeled data).

Table 6: Evaluation of outlier detection on outliers unseen in unlabeled training data (AUROC). Higher is better. Supervised models use the same batch size, learning rate as OpenMatch, but are trained with fully labeled inliers.

**Limitations.** OpenMatch can have difficulty in detecting outliers similar to inliers, though any methods for outlier detection can have the same limitation. If the outliers have similar visual characteristics, separating them from inliers samples is difficult. A potential solution is to introduce self-supervised learning [15, 4] for unlabeled data.

**Broader Impact.** OpenMatch is an effective tool for semi-supervised learning with noisy unlabeled data containing outliers. It would benefit projects with small budgets or hard-to-label data. At the same time, it is possible for malicious actors to exploit the advantages of learning from limited data. Broadly speaking, any progress on semi-supervised learning can have these consequences.

# 6 Acknowledgement

This work was supported by Honda, DARPA LwLL and NSF Award No. 1535797.

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
