# A  Experimental Details

## A.1  Implementation

We utilize `https://github.com/kekmodel/FixMatch-pytorch` to implement Labeled Only, OpenMatch and FixMatch. The implementation reproduces the result of FixMatch well. For MTC, we employ the author's official implementation with Pytorch.

**Hyper-parameters.** Unless otherwise noted, the same value is used for all experiments.

- The batch-size $B = 64$.
- The relative size of batch-size for unlabeled data $\mu = 2$.
- Trade-off parameters $\lambda_{em} = 0.1$, $\lambda_{fm} = 1$.
- Trade-off parameter $\lambda_{oc} = 0.5$ for CIFAR10 and ImageNet, $\lambda_{oc} = 1.0$ for CIFAR100.
- Epochs to begin FixMatch training, $E_{fix} = 10$.
- Iterations per epoch $I_{max} = 1024$.
- Total number of epochs $E_{max} = 512$.
- Optimizer: SGD with nesterov momentum = 0.9.
- Learning rate $\eta = 0.03$.

Hyper-parameters shared with FixMatch are borrowed from the implementation except for that $\mu$ and $E_{max}$ are set smaller to reduce computation time. To make a fair comparison with FixMatch, we utilize hyper-parameters defined above to train a FixMatch model.

**Validation.** In CIFAR10 and CIFAR100, 50 samples per each known class are used. For ImageNet-30, 10% of the training split is used for the purpose.

**MTC.** Since the author provides implementation optimized for CIFAR10, we first employ the same hyper-parameters. Although we attempted to tune the hyper-parameter such as a trade-off weight for out-lier detection loss, the results did not improve with the tuning. Therefore, we use the same hyper-parameters for all experiments.

## A.2  Dataset

**ImageNet-30.** We utilize the training split of ImageNet-30 for training and test one for evaluation. To be specific, 2,600 samples are employed for labeled training data and validation respectively, 33,800 samples are employed for unlabeled data. Testing was done for 3,000 test samples.

**Novelty Detection Datasets** We follow CSI [1] to set up the datasets. See their implementation `https://github.com/alinlab/CSI` for more details.

# B  Additional Results

We provide results of the ablation study and hyper-parameter sensitivity analysis.

| SOCR | CIFAR10-50 | CIFAR10-400 |
|---|---|---|
|  | 14.6±2.1 / 49.5±15.2 | 9.9±1.3 / 86.3±5.2 |
| ✓ | **10.4±0.9 / 99.3±0.3** | **5.9 ±0.5 / 99.3±0.2** |

Table A: Ablation study (Error rate / AUROC). Adding SOCR clearly boosts the performance in both metrics.

**Additional ablation study.** In the ablation study of the main paper, we also ablate FixMatch loss for clarity of ablation. Here, we show the ablation of SOCR for a model training with FixMatch loss. In Table A, SOCR improves the performance with a large margin.

**Ablation study for entropy minimization.**

| Ent | SOCR | CIFAR10-50 | CIFAR100-50 |
|:---:|:----:|:----------:|:-----------:|
| ✓ | | $60.5 \pm 2.8$ | $70.4 \pm 0.1$ |
| | ✓ | $78.1 \pm 1.9$ | $78.7 \pm 0.1$ |
| ✓ | ✓ | $\mathbf{81.3 \pm 2.9}$ | $\mathbf{78.9 \pm 0.1}$ |

Table B: Ablation study with entropy minimization (AUROC). Adding entropy minimization slightly improves AUROC.

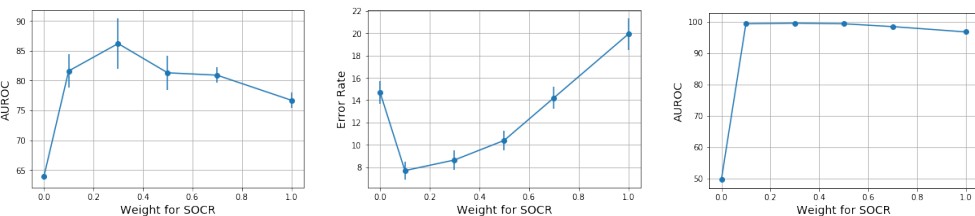

(a) AUROC without FixMatch          (b) Error rate with FixMatch          (c) AUROC with FixMatch

Figure A: Sensitivity analysis with respect to the weight for SOCR loss. (a): AUROC of models trained without FixMatch. (b): Error rate of models trained with FixMatch. (c): AUROC of models trained with FixMatch.

In Table B, we provide the ablation study including entropy minimization by the outlier detector. We can see that combining entropy minimization slightly improves performance, but most of the gain is from SOCR.

**Sensitivity to hyper-parameters.**

Fig. A illustrates the sensitivity to the weight parameter for SOCR. Fig. A(a) describes the case where FixMatch loss is not employed. Although there is a small variation in AUROC by the value of weight parameters, superiority over a model without SOCR (weight equal to 0) is clear in all cases. In Fig. A(b) and (c), error rate and corresponding AUROC by varying weight parameter while fixing other hyper-parameters are shown. We observe that increasing the weight decreases the number of pseudo-inliers. Since we do not change the threshold in this experiment, the number of unlabeled inliers used for FixMatch decreased, which results in a larger error rate. On the other hand, AUROC remains high in every weight value, which indicates the effectiveness of SOCR to separate inliers and outliers. Also, when SOCR is not employed for training (weight is 0), AUROC significantly degrades.

# References

[1] Jihoon Tack, Sangwoo Mo, Jongheon Jeong, and Jinwoo Shin. Csi: Novelty detection via contrastive learning on distributionally shifted instances. *arXiv preprint arXiv:2007.08176*, 2020.