# OpenReview forum: "OpenMatch: Open-Set Semi-supervised Learning with Open-set Consistency Regularization"
_NeurIPS.cc/2021/Conference — NeurIPS 2021 Poster_

### Official Review · Reviewer_6PWa · 2021-06-28

**Rating:** 5
**Confidence:** 5

**Summary:**

This paper focuses on semi-supervised learning with outlier unlabeled data. This problem has attracted much attention in the semi-supervised learning community. The authors propose a method that contains two parts: First, detect outliers based on the OVA classifier. Second, propose an open-set soft-consistency regularization loss that can help improve the outlier detection performance.

**Limitations And Societal Impact:**

Yes.

**Main Review:**

This paper focuses on semi-supervised learning with outliers, more accurately, the unlabeled data contains some examples that belong to classes that unseen in labeled data.  The authors propose an outlier detection-based method to detect outliers and help improve the SSL performance. I have the following concerns about this paper:

1) The contribution is limited. The proposal is mainly based on an existing outlier detection method. It seems that if we can detect outliers accurately and delete these outlier examples, the problem becomes an ideal SSL setting. So the open-set semi-supervised learning problem is actually an outlier detection problem?
2) Based on question 1, in this paper, the authors do not compare the proposal with the SOTA OOD detection method. The performance of the combination of OOD detection methods and SSL methods should be reported.
3) What are the differences between outlier detection and open-set learning. In my view, open-set learning should make the model has the ability to classify the new class data.
4) Self-Supervised learning, e.g., SimCLR, can help to learn a separatable representation and help OOD detection tasks. There are many OOD detection algorithms based on contrastive learning that have been proposed. Can we simply pre-train a model with SimCLR using all labeled and unlabeled data, and based on the learned representation to detect outlier unlabeled examples. Have you tried this way in your experiments?

**Time Spent Reviewing:**

0.6 hours

---

> ### Author Response · Authors · 2021-08-09
> **Contribution and comparison with OOD methods.**
>
> Thanks for your helpful feedback. We will answer the concerns in the following.
>
> ### Contribution.
> The open-set semi-supervised learning (OSSL) is different from the outlier detection problem in two points. (1) The amount of provided labeled data is limited in open-set semi-supervised learning, while outlier detection works use many labeled inlier samples. (2) We have unlabeled data that includes both known and unknown classes, but we do not know which are known or unknown. By contrast, in OOD (multi-class classification case in [36]), the training set consists of only clean inlier data but does not contain outliers (E.g., trained only on CIFAR-10 and tested on outlier datasets). While OOD aims to detect samples of datasets unseen during training, the model needs to distinguish between outliers and inliers in unlabeled training data.
> With regard to the novelty, note that we do not simply apply existing outlier detection techniques for open-set semi-supervised learning. We propose to improve the performance in open-set semi-supervised learning by introducing a consistency loss w.r.t outlier detector, which shows significant improvement in outlier detection. Therefore, our work does not simply apply the outlier detection technique for open-set semi-supervised learning.
>
> ### Comparison with OOD detection method.
>
> As we answer the question above, the settings of OSSL and OOD are quite different. The recent state-of-the-art techniques in OOD are applying self-supervised learning loss for labeled data consisting of only *inliners*, while our unlabeled training data contains both inlier and outlier.
>
> But, please note that our focus is to utilize unlabeled data consisting of both inliers and outliers and we propose a new objective to improve outlier detection. In this sense, we think that the contributions of our work and existing OOD are not overlapping.
>
> In addition, we believe that if the OOD methods for labeled data are helpful in OSSL, we can add our objective for unlabeled data on top of the OOD methods.
>
> Also, please refer to our response to Reviewer Gu7C ("Contrastive learning is helpful?"). Contrastive learning is one of the state-of-the-art methods in OOD detection ([36], [40]). We provide the results and observe that introducing SimCLR for unlabeled data or initializing a model pre-trained with SimCLR does not improve the performance.
>
> ### Differences between outlier detection and open-set learning.
> The goal of OOD and open-set learning is quite similar. In our understanding, open-set learning often focuses on multi-known-class classification whereas OOD handles one-class classification too. Both models are required to identify samples of new classes as a novel class.
>
> ### Self-Supervised learning.
> Please refer to our response to Reviewer Gu7C ("Contrastive learning is helpful?"). We provide results using SimCLR. But, we could not observe a clear improvement by applying SimCLR to unlabeled data or initializing a model pre-trained with SimCLR.

---

> > ### Comment · Reviewer_6PWa · 2021-08-13
> > **Reply**
> >
> > In fact, not all OOD detection algorithms assume there are many labeled examples, there are also unsupervised OOD detection algorithms, and there are also some unsupervised OOD detection methods that assume unlabeled data include both known and unknown classes. For example, the case studied in paper [1] is very similar to this paper. If I understand correctly, the OSSL setting assumes there is a small set of labeled data that includes all known classes and a large amount of unlabeled data that includes both known classes and unknown classes. In my view, we can simply adopt an unsupervised OOD detection method or an unsupervised representation learning method without using labels, and based on the label set to select all examples that don't belong to known classes. Then the problem becomes an ideal SSL setting. This method is very straightforward. So, in my view, the problem considered in this paper is not novel.
> >
> > [1] Yu, Qing, and Kiyoharu Aizawa. "Unsupervised out-of-distribution detection by maximum classifier discrepancy." Proceedings of the IEEE/CVF International Conference on Computer Vision. 2019.

---

> > > ### Author Response · Authors · 2021-08-14
> > > **Reply**
> > >
> > > Thanks for the question.
> > >
> > > "in my view, the problem considered in this paper is not novel." => We do not claim that the open-set semi-supervised learning is the novel setting we propose. We are citing and comparing with papers that tackled the same problem [44, 6, 14].
> > >
> > > "In my view, we can simply adopt an unsupervised OOD detection method or an unsupervised representation learning method without using labels, and based on the label set to select all examples that don't belong to known classes." =>
> > > We would like to explain why we think simply adopting an unsupervised OOD detection method will not easily solve the OSSL. By "simply adopting OOD", we mean a method that performs outlier detection to pick inliers from unlabeled data and applies semi-supervised loss to them.
> > >
> > > Even if we are provided with many labeled or unlabeled inliers, detecting all samples that are out of the training dataset is very hard, as existing OOD work indicates. Since the number of inliers (labeled samples) in OSSL is much smaller than OOD methods are using, detecting outliers is even harder. Therefore, simply adopting unsupervised OOD detection will not distinguish inliers and outliers enough. Note that unlabeled data is a mixture of inliers and outliers.
> > > Our consistency loss helps with this problem by enhancing the separation between inliers and outliers within unlabeled data, and is shown to perform well to separate them.
> > >
> > > From the reason above, OSSL is different from OOD, and simply applying OOD techniques will not be enough to solve the problem. We are showing several results in the response to Reviewer YUGo, in which contrastive learning was not helpful.
> > >
> > > Additionally, we would like to highlight that our proposed method and previous work MTC [44] have the form of unifying the OOD method and SSL to handle OSSL effectively. A clear difference from "simply adopting OOD" is that two approaches employ an objective for unlabeled data in order to separate inliers and outliers.
> > >
> > > We utilize the OVA-classifier as an outlier detector and apply FixMatch for SSL. Samples treated as inliers by the outlier detector are fed into the network to compute the SSL loss.
> > > To enhance the separation between inliers and outliers, we propose to utilize consistency loss for unlabeled samples. We understand that performing outlier detection to pick inliers from unlabeled data and applies semi-supervised loss to them can be a baseline in OSSL.
> > > Ablating SOCR from our approach is this kind of baseline.
> > > As the response to Gu7C indicates, this baseline gives poorer performance than ours. This fact demonstrates the need to incorporate the loss to distinguish unlabeled inliers and outliers, which is not introduced for a standard OOD setting due to the difference in problem setting.
> > >
> > > From these discussions, we believe that the problem of OOD and our OSSL has substantial differences.

---

> > > > ### Comment · Reviewer_6PWa · 2021-09-03
> > > > **Experiments on Novelty Detection**
> > > >
> > > > Results in Table 4 are not clear. What is the meaning of the first column? The AUCROC is adopted to evaluate the ability to separate inliers and outliers. If I understand correctly, the first column on CIFAR10 means we only use the CIFAR10 dataset without the OOD dataset. How to compute the AUCROC in this case?

---

> > > > > ### Author Response · Authors · 2021-09-03
> > > > > **Response to a question on novelty detection**
> > > > >
> > > > > Thanks for the question.
> > > > >
> > > > > In the first column of Table 4, we are showing the AUROC to separate inliers and outliers within the dataset used for training. For example, inliers and outliers of CIFAR10 are employed to compute the AUROC in the first column. Please note that the values in the first column of Table 4 and the corresponding values in Table 2 are the same.
> > > > > We will clarify this point in the caption in a camera-ready paper.

---

### Official Review · Reviewer_5gr3 · 2021-07-03

**Rating:** 6
**Confidence:** 4

**Summary:**

This paper tackles open-set semi-supervised learning (OSSL) where outliers exist in unlabeled data. The key idea for detecting outliers is to learn class-specific, one-vs-all (OvA) classifiers—if all classifiers produce negative outputs for an unlabeled data, the data is considered an outlier. In addition to the classification loss computed on labeled data, the OvA classifiers are also trained on unlabeled data with an entropy minimization loss and a consistency regularization loss—the latter is computed on two views of the same data. The results on CIFAR and ImageNet show that the proposed approach largely outperforms the baselines in terms of both recognition accuracy and outlier detection accuracy.

**Limitations And Societal Impact:**

The authors have properly covered these aspects in Sec.5.

**Main Review:**

**Strength**
- The design of combining FixMatch with the OvA classifiers is simple and effective, which could add value to the research of OSSL.
- The experiments are convincing and extensive, covering both classification and outlier detection tasks.
- The improvements over previous SOTA are impressive.
- The paper is easy to follow.

**Weakness/Questions**
- I am a bit concerned with the novelty. The idea of using the OvA classifiers for detecting outliers is borrowed from [32], which has been acknowledged in this paper. [32] deals with the problem of open-set unsupervised domain adaptation (UDA). Given the closedness between UDA and SSL, it's not too surprised to see that the OvA classifiers work in the OSSL setting. One could argue that UDA and SSL are different problems since the former needs to cope with domain shift while the latter doesn't, but we couldn't ignore many recent studies that have shown that methods developed for one problem (either UDA or SSL) work very well on the other. Nonetheless, I still appreciate the simplicity and effectiveness of the proposed approach.
- Why compute the consistency loss on logits instead of the output probabilities? Is it because the former gives a better result?
- Eq.1 looks a bit weird as it is supposed to be the binary cross-entropy loss.
- The entropy minimization loss $L_{em}$ defined in Eq.2 is not evaluated in the ablation study. One would be wondering if the improvement brought by this loss is significant.
- The motivation for SOCR is not very convincing. This loss is computed for the sub-classifiers. Why this loss could help the closed-set classifier produce more accurate pseudo-labels?

---
Post-rebuttal update:

I have read the authors' responses and other reviewers' comments as well. I keep my rating unchanged.

**Time Spent Reviewing:**

2.5

---

> ### Author Response · Authors · 2021-08-09
> **Ablation for entropy minimization.**
>
> Thanks for your helpful feedback. We will address the concerns below.
>
> ### Novelty.
> Please refer to our response to Reviewer Gu7C ("Novelty").
>
> ### The consistency loss on logits instead of the output probabilities.
> We compute the loss on output probabilities of the outlier detector.
>
> ### Eq. 1
> OVANet formulated the loss as in Eq. 1. They employed softmax activation over N $\times$ 2 vectors, where N is the number of samples in mini-batch.
>
> ### Ablation on entropy minimization loss.
> Please refer to our response to Reviewer YUGo ("The ablation study to turn off SOCR").
>
> ### The motivation for SOCR.
> First, since the loss from the OVA sub-classifiers is propagated to a feature extractor shared with a closed-set classifier, the feature extractor can provide features separating known and unknown instances well, which can avoid giving unknown instances wrong pseudo-labels. Second, selecting pseudo-inliers (Line7 in Alg. 1) can work better with the OVA sub-classifiers trained with SOCR.

---

> > ### Comment · Reviewer_5gr3 · 2021-08-11
> > **Follow-up questions**
> >
> > > We compute the loss on output probabilities of the outlier detector.
> >
> > Line 51, "we minimize the distance between the two logits". Perhaps it's a typo?

---

> > > ### Author Response · Authors · 2021-08-11
> > > **Response to follow-up**
> > >
> > > Yes. We will fix the issue in camera-ready. Thanks for pointing it out.

---

### Official Review · Reviewer_Gu7C · 2021-07-15

**Rating:** 6
**Confidence:** 3

**Summary:**

The paper proposes a model for open-set semi-supervised learning. As a semi-supervised learning model, it tackles classification where only a portion of the training data is labeled. "Open set" refers to the fact that the unlabeled data are noisy and can contain out of distribution examples, for which the class is not among the known classes. Out of distribution samples are present also during test, and the model has to recognize them and avoid classification on them. The proposed model builds on top of existing works, combining several loss terms and ideas such as one-versus-all classifiers and FixMatch as the main semi-supervised learning engine. The main technical contribution is a loss term enforcing the out-of-distribution score for two augmented versions of the same unlabeled example to be similar. Extensive experiments validate the proposal against recent state-of-the-art models.

**Main Review:**

This submission has some solid merits, as it tackles a meaningful problem for the machine learning community, namely training in the presence of noisy labels and out of distribution examples.
The method is straightforward and embeds a combination of several existing components, such as:
- FixMatch as the main semi-supervised learning engine. As is, it cannot deal well with the open set scenario.
- OVA-classifiers to identify in-distribution and out-of-distribution unlabeled examples, and therefore pseudo-label the formers and reject the latters. Originally, OVA-classifiers were proposed (in the version used here) for universal domain adaptation.
This results in the biggest weakness of the paper: the limited technical contribution. Indeed, the only source of novelty is in a regularization loss (Eq. 3) that enforces, on an unlabeled sample, the out-of-distribution scores of two augmented versions to be closer.

Following on the weak points, the model involves a number of hyperparameters, specifically in weighting the loss function terms. An analysis about sensitivity of performances to those lacks, and it is a bit unclear how one should select their value.

Clarification: in Tab. 2, how do the baseline "Labeled Only" and "FixMatch" perform OOD detection? Are they equipped with OVA classifiers as well? In their original formulation no mechanism allows for OOD. Is it with confidence thresholding?

I consider the ablation study on the proposed open-set consistency regularization loss particularily important, as it validates the main novel contribution of the paper. The benefit of the proposed loss function is demonstrated in Tab. 3, but I find curious that the authors turned off another loss component for this experiment (FixMatch). What is the reason for that? The text mentions "to measure the pure gain from the consistency loss", it is unclear to me. I would find ablating the consistency loss, whilst activating all other components, much cleaner.

Despite these limitations, the experimental section is solid and I think it is the true strength of the paper.
The experimental protocols used seem correct, and tests are carried out on 3 different datasets at different degrees of inlier-outlier splits. Performances of the model are properly reported both in terms of classification accuracy on the known classes and outlier detection. On both tasks, the proposed OpenMatch model outperforms recent competing methods by a significant margin.

---

Overall, I consider this a borderline submission. The weakest point is the minor technical contribution, as the main novel part is the consistency loss and other components are off-the-shelf modules. The strongest point is experimental evaluation, that shows big performance gap with respect to very recent baselines. All things considered, I lean more towards accepting the paper.

**Time Spent Reviewing:**

4

---

> ### Author Response · Authors · 2021-08-09
> **Novelty and ablation study for SOCR.**
>
> Thanks for reviewing our paper. We will address the concerns.
>
>
> ### Novelty.
> It is true that an OVA-classifier is proposed by a previous work. But, the combination of consistency loss and OVA-classifier is new. In addition, we would like to highlight that our loss is simple, yet produces a large gain in OOD detection in OSSL. We believe this is an important fact worth sharing in the research community of semi-supervised learning and outlier detection.
>
> ### Clarification.
> We train the OVA-classifier on top of the two models.
>
> ### Hyperparameters.
> We choose the hyper-parameters based on the accuracy of validation samples as stated in the line 207. The accuracy on the validation samples is correlated with AUROC and accuracy on the test set well. We will add the analysis on sensitivity to the weight term in camera ready.
>
> ### The ablation study of turning off SOCR.
>
> We observe that the use of FixMatch can improve or decrease the performance of AUROC (Table 2).
> To ablate the gain or decrease from FixMatch, we turned off the loss in Table 3.
>
> Here, we conducted ablation experiments for SOCR loss while turning on FixMatch.
>
> ##### Results of ablation for SOCR
>
> | Error | CIFAR10-50 | CIFAR10-400 |
> | -------- | -------- | -------- |
> | w/o SOCR     | 14.6 $\pm$ 2.1    | 9.9 $\pm$ 1.3|
> | w SOCR     | 10.4 $\pm$ 0.9     | 5.9 $\pm$ 0.5|
>
> | AUROC | CIFAR10-50 | CIFAR10-400 |
> | -------- | -------- | -------- |
> | w/o SOCR     | 49.5 $\pm$ 15.2 |  86.3 $\pm$ 5.2   |
> | w SOCR     |99.3 $\pm$ 0.3    | 99.3 $\pm$ 0.2|
>
> The improvements by SOCR are clear, especially in AUROC. Since SOCR improves the performance to find outliers, we can select pseudo-inliers effectively.
> We will add the discussion in our camera ready.

---

> > ### Comment · Reviewer_Gu7C · 2021-09-02
> > **Response**
> >
> > I thank the authors for the clear response they provided.
> >
> > I appreciate the new ablation study showing that SOCR has a high impact on outlier detection and in turn to the global performance of the model.
> >
> > I will keep my score unchanged in light of the limited technical contribution (as also highlighted by other referees). However, I consider the experimental section solid and overall the merits outnumber the weaknesses. Thus, I confirm that I lean towards acceptance.
> >
> > Best,
> > Gu7C

---

### Official Review · Reviewer_YUGo · 2021-07-16

**Rating:** 7
**Confidence:** 4

**Summary:**

The authors propose an idea to improve model's performance under the open-set semi-supervised learning setting. This work proposed to leverage OOD detector, combined with FixMatch, and soft open-set consistency regularization (SOCR) to improve the model training during open-set semi-supervised learning. I think the usage of OOD on ignore outliers is not new and an unsupervised method on learning the consistency between augmentations for unlabeled data is also not new. Thus, I am not sure if I can find too much of novelty from this work itself, but I do appreciate the author's effort in putting them together to be a complete model, which in fact make solid progress on the open-set semi-supervised learning setting.

**Ethical Concerns:**

I think one potential issue for semi-supervised learning is always that the model could be biased since we are leveraging a smaller number of samples that could already be biased. This essentially is what most of the semi-supervised learning papers tried to address, yet I don't think any of us actually achieved that yet.

**Limitations And Societal Impact:**

Yes, the authors provided the "Limitations" and "Broader Impact" subsections in the conclusion.

**Main Review:**

### Strengths: Describe the strengths of the work. Typical criteria include: soundness of the claims (theoretical grounding, empirical evaluation), significance and novelty of the contribution, and relevance to the NeurIPS community.

- A solid advance on a practical Semi-Sup. Learning setting, though there isn't too much novelty in the proposed method.
- Setting Benchmark by considering MTC, UASD, and D3SL as baselines.
- Good experimental results and analysis

### Weaknesses: Explain the limitations of this work along the same axes as above.

- The related work section is hard to follow. I think it lacks proof-read, and its writing quality is much worse than the introduction section.
- Fig. 2 is quite confusing. Why are the y-axis of (b) and (c) for "P of outliers for samples in circle shape"? Aren't the outliers samples in diamond shape? Also, I am not sure if I follow (C), in which the authors try to depict what the proposed soft consistency will do to the predictions of labeled inliers and unlabeled outliers. I think what Soft Open-set Consistency Regularization (SOCR) is trying to do is basically forcing the model to learn better visual representations of the unlabeled outliers. I would be curious what's the difference here if we were to apply contrastive learning to the unlabeled outliers.

- I don't think the authors ablated the Entropy Minimization. How important is this component in the proposed work?

### Please provide an "overall score" for this submission.

Overall, I think this paper made a solid contribution to the semi-supervised learning community. I am less satisfied with the novelty of the paper, though I don't think this necessarily leads to grounds for rejection. Personally, I believe that any work that makes meaningful and solid progress for a practical research problem is worth to be shared with the research community. That being said, I do suggest the authors do a proofread on their paper, especially revise some of the confusing parts in the paper.

**Time Spent Reviewing:**

6

---

> ### Author Response · Authors · 2021-08-09
> **Contrastive learning and ablation study for entropy minimization.**
>
> Thank you for carefully reading our paper. We will address the concerns in the following.
>
> ### Related work.
> We will improve our writing. Thanks for pointing it out.
>
> ### Fig. 2
> Since the y-axis shows the probability of being an outlier for the one-vs-all outlier detector of the circle class, the shape of the symbol is correct. But, we will try to clarify this point in our camera ready.
>
> ### Initializing a model with Contrastive learning.
> Reviewer YUGo and 6PWa asked whether initializing a model with contrastive learning helps or not.
> Given the question, we performed two additional experiments: (1) initializing a model pre-trained with SimCLR, (2) incorporating SimCLR loss during training.
>
> In summary, we could not find clear gains from SimCLR.
> SimCLR loss tries to spread all samples in a unit sphere while the goal of OSSL is to cluster inliers into the correct classes and to separate outliers from inliers. Incorporating SimCLR loss does not necessarily have this effect.
>
> Note that the experimental setting here is the same as Table 1 and 2.
> #### Results of experiments (1)
>
> | Error | CIFAR10-50 | CIFAR10-400 |CIFAR100-50|
> | -------- | -------- | -------- |-------- |
> | SimCLR     | 10.6 $\pm$ 0.1     | 5.9 $\pm$ 0.3    |27.1 $\pm$ 0.5|
> | Random | 10.4  $\pm$ 0.9     | 5.9 $\pm$ 0.5     |27.7 $\pm$ 0.4  |
>
> | AUROC | CIFAR10-50 | CIFAR10-400 |CIFAR100-50|
> | -------- | -------- | -------- |-------- |
> | SimCLR     | 99.3 $\pm$ 0.1     | 99.4 $\pm$ 0.1     |87.1 $\pm$ 0.8|
> | Random     |   99.3 $\pm$ 0.3  |99.3 $\pm$ 0.2    |87.0 $\pm$ 1.1|
>
> We can see that there is no clear difference between the random initialization (Random) and SimCLR initialization (SimCLR).
>
> ####  Results of experiments (2)
>
> Next, instead of initializing a model with SimCLR, we train a model with supervised loss and SimCLR loss.
> We test two cases: (1) SimCLR for labeled data [40] and (2) SimCLR for unlabeled data.
> Note that the experimental setting here is the same as Table 3, where we turn off FixMatch.
>
>
> | AUROC  | CIFAR10-50 | CIFAR100-50 |
> |-------- | -------- | -------- |
> |Labeled Only |  63.9 $\pm$ 0.5 | 70.3 $\pm$ 0.5|
> |SimCLR for Labeled|   63.5 $\pm$ 0.7    | 68.6 $\pm$ 1.1 |
> |SimCLR for Unlabeled|  62.1 $\pm$ 1.2   |  68.8 $\pm$ 0.9 |
> | Ours | 81.3 $\pm$ 2.9   | 78.9 $\pm$ 0.1    |
>
> Adding SimCLR loss harmed the performance. Here, We set the trade-off weight for SimCLR as 0.1.
> Note that we tried different trade-off weights for SimCLR loss, but could not see any gain by incorporating the loss. We will include these experiments in camera-ready.
>
> ### Ablation for Entropy Minimization.
>
> We added an ablation study for entropy minimization (ENT) as in Table 3. The table below shows AUROC.
> In summary, the gain from SOCR is much larger than the one from ENT. Having SOCR loss allows a model to effectively separate inliers from outliers.
>
> #### Results of ablation for entropy minimization
>
> |ENT | SOCR                  | CIFAR10-50 | CIFAR100-50 |
> |--------|  -------- | -------- | -------- |
> |$\checkmark$|   | 60.5 $\pm$ 2.8| 70.4 $\pm$ 0.1|
> |    |$\checkmark$           | 78.1 $\pm$ 1.9      | 78.7   $\pm$ 0.1  |
> |$\checkmark$ |$\checkmark$  | 81.3 $\pm$ 2.9   | 78.9 $\pm$ 0.1    |

---

> > ### Comment · Reviewer_YUGo · 2021-09-02
> > **Thank you for running the experiments for further ablation study**
> >
> > Hi authors,
> >
> > Thank you so much for running the experiments with my request. It would be great if the authors will be able to include the additional experiments in the revision if the paper was accepted afterwards.
> >
> > My rating for the paper remains the same. I still think the proposed experimental setting, experimental results and analysis, and insights that come with the paper are still worth to be shared with the research community, despite slight set back on the novelty of the method. Thus, my final rating remain the same. I am leaning towards acceptance for this paper.

---

### Official Review · Reviewer_fHpH · 2021-08-27

**Rating:** 6
**Confidence:** 4

**Summary:**

This manuscript introduces OpenMatch for open-set semi-supervised learning. The OpenMatch integrates a one-vs-all classifier (OVA-classifier, working as an outlier detector) and FixMatch. The main contribution should be the soft open-set consistency regularization (SOCR) in Figure 1 and the OpenMatch Framework in Alg 1. Specifically, the SOCR follows the same self-supervised framework in FixMatch but it uses OVA-classifier to output consistent anomaly score distribution. After using the model in Figure 1 to identify inliers from unlabeled data, the FixMatch is allowed to reach a higher accuracy on the OSSL tasks.

**Limitations And Societal Impact:**

yes.

**Main Review:**

My main concern is about the experiments.

1)The studied problem of open-set semi-supervised learning (OSSL) is close to the problem of out-of-distribution (OOD) detection.
And there are some related works also considering semi-supervised settings for OOD:
- 2020 Robust Semi-Supervised Learning with Out of Distribution Data
- 2020 AAAI Semi-Supervised Learning under Class Distribution Mismatch

However, in experiments, only two baselines FixMatch (not OOD method) and MTC are mainly considered. This may not be convincing enough to demonstrate its superiority.
Thus, more baselines (especially related OOD methods) are encouraged to include.

2)For the model in Figure 1, it should be an *unsupervised* OOD classifier. As the soft open-set consistency regularization (SOCR) is one of the main contributions in this work, it would be better to compare the figure 1 model with the main-stream OOD methods (as they can also be used to identify OOD samples).

Overall, this paper is well written and the provided solution of using OVA-classifier to help FixMatch is well motivated. The authors demonstarted that using OVA-classifier with SOCR can help FixMatch on open-set semi-supervised learning (OSSL). However, related OOD baselines are not included in comparisons.

**Time Spent Reviewing:**

8

---

> ### Author Response · Authors · 2021-09-01
> **Response to a review**
>
> Thanks for your feedback.
> We would like to address your concerns.
>
> 1. As a comparison to "2020 AAAI Semi-Supervised Learning under Class Distribution Mismatch", our model outperforms their method by a large margin. In Fig.3 of their paper, the error rate in cifar10-400 (400 labeled samples per class) was at least 20% while ours shows 5.9% in the setting. Note that they are using the same backbone and the same class splits.
> 2. "2020 Robust Semi-Supervised Learning with Out of Distribution Data" is a concurrent work, which we did not choose to compare.
> 3. We compare our method with contrastive learning [40], which is one of the popular approaches in OOD. See response to YUGo. But, we did not see a clear gain from using contrastive learning in this task.

---

### Decision · Program_Chairs · 2021-09-27

**Decision:**

Accept (Poster)

**Comment:**

The paper proposes a model for open-set semi-supervised learning. It is well-written and well-motivated. As pointed out by almost all the reviewers, there is limited novelty as the proposed model is a straightforward combination of existing components. However, extensive experimental evaluation shows big performance gap with respect to very recent baselines, and shows solid progress on the open-set semi-supervised learning setting studied.

To further make the experimental results more convincing, it will be useful to add sensitivity analysis on the hyperparameters (as the model involves a number of them). Also, as the problem is related to OOD methods, the authors are encouraged to include OOD-related baselines.